# Biologic Mechanisms of Macrophage Phenotypes Responding to Infection and the Novel Therapies to Moderate Inflammation

**DOI:** 10.3390/ijms24098358

**Published:** 2023-05-06

**Authors:** Renhao Ni, Lingjing Jiang, Chaohai Zhang, Mujie Liu, Yang Luo, Zeming Hu, Xianbo Mou, Yabin Zhu

**Affiliations:** Health Science Center, Ningbo University, Ningbo 315211, China; 206002240@nbu.edu.cn (R.N.); 206004206@nbu.edu.cn (C.Z.); mouxianbo@nbu.edu.cn (X.M.)

**Keywords:** inflammation, macrophage phenotype, cellular signal, biologic mechanism

## Abstract

Pro-inflammatory and anti-inflammatory types are the main phenotypes of the macrophage, which are commonly notified as M1 and M2, respectively. The alteration of macrophage phenotypes and the progression of inflammation are intimately associated; both phenotypes usually coexist throughout the whole inflammation stage, involving the transduction of intracellular signals and the secretion of extracellular cytokines. This paper aims to address the interaction of macrophages and surrounding cells and tissues with inflammation-related diseases and clarify the crosstalk of signal pathways relevant to the phenotypic metamorphosis of macrophages. On these bases, some novel therapeutic methods are proposed for regulating inflammation through monitoring the transition of macrophage phenotypes so as to prevent the negative effects of antibiotic drugs utilized in the long term in the clinic. This information will be quite beneficial for the diagnosis and treatment of inflammation-related diseases like pneumonia and other disorders involving macrophages.

## 1. Introduction

Inflammation serves as a defense for the body against both external and internal injuries. Its main function is to maintain tissue homeostasis by facilitating the transport of nutrients and immune cells through the blood. However, inflammation is a complex process encompassing various responses, including tissue injury, immune defense, and tissue repair [1]. During the inflammatory stage, the differentiation and migration of numerous types of cells are closely interconnected, and the disruption of the inflammatory process may lead to a range of severe diseases, such as ulcers, sepsis, myocardial infarction, and pneumonia, particularly viral pneumonia caused by the COVID-19 epidemic rooting from the SARS-CoV-2 virus [2,3], while disease like monkeypox (mpox) virus infection is more likely to occur in immunocompromised individuals. In this case, macrophages are the main targets of mpox, along with the epithelial cells, plasmocytes, fibroblasts, and dendritic cells in the mpox-infected tissues/organs [4]. The immune system will undergo harm as a result of the lethal mpox’s immunosuppression, thereby increasing the likelihood of pathogen invasion into the human body. According to the literature [5,6], mpox-infection will continually attack CD4^+^ T cells and macrophages, making the patient ultimately die from severe skin lesions, lung involvement, secondary infections, and sepsis. Thus, the correlation between inflammation and pathogenesis has been a prominent research topic in medical science in the past several decades. 

The innate immune system plays a crucial function in initiating adaptive immune responses. Macrophages, as the signaling hub of the immune system, play a significant role in presenting antigens, promoting the proliferation and differentiation of lymphocytes, and activating immune responses to regulate human innate immunity [7]. In addition, macrophages can differentiate into pro- and anti-inflammatory phenotypes in response to external stimuli, thereby playing a critical role in the development or suppression of inflammation.

Due to their various roles and polymorphisms in regulating immune responses and metabolism, macrophages have been closely linked to the progression of several diseases [8,9]. In the infected tissues, macrophages initially polarize into a pro-inflammatory M1 phenotype, releasing inflammation-related mediators that recruit other inflammatory cells, like neutrophils or/and leukomonocytes, as a response to fight the pathogens. Following the resolution of inflammation, macrophages with the M1 phenotype will transition into M2 phenotype to assist in the repair of damaged tissues [10,11]. However, in the progression of pneumonia, cytokine storms caused by excessive responses of M1-type macrophages and the secretion of excessive cytokines such as tumor necrosis factor-alpha (TNF-α), interleukin-1 beta (IL-1β), interleukin-6 (IL-6), and other factors can be fatal [3,12]. It will lead to death if the inflammation is not handled properly. 

Since an immune system disbalance would result in such a catastrophic event in our body, it is necessary to concentrate our research on discovering novel strategies to identify inflammation and mitigate the inflammatory seriousness or to offer techniques to treat inflammation-related diseases. Thus, we reviewed the references for the inflammation diagnosis and the achievements regarding therapeutic biomaterials that take effect in the phenotypic conversion of macrophages, attempting to elucidate the biological mechanisms of the macrophages in regulating inflammation and participating in tissue regeneration.

## 2. Significance of Anti-Inflammation in Disease Treatments

Inflammation is an adaptive response of the body to jeopardous stimuli. It is able to occur in all tissues and organs of the body, though generally confined to a specific site. Inflammation is the dynamic progress of injury, anti-injury, and tissue repair [13]. In the case of infection by pathogenic microorganisms, the inflammation will spread from the primary site to the whole body, resulting in a systemic inflammatory response. Leukocytes and mononuclear phagocytes quickly begin proliferating during a virus invasion or bacterial infection and produce chemokines, cytokines, vasoactive peptides, or other substances to transfer to the inflammatory site via blood vessels. Subsequently, macrophages and dendritic cells are responsible for presenting antigens to lymphocytes, promoting their differentiation and activating cellular and humoral immunity, aiming at controlling the inflammation in the local tissues. The majority of inflammatory responses are moderate and beneficial. The immune system will transition into an anti-inflammatory phase once the inflammatory agents have been eliminated. Pro-inflammatory macrophages and neutrophils are replaced by restorative immune cells like fibroblasts and anti-inflammatory macrophages, which reshape the extracellular matrix and promote the regeneration of the injured tissue [14]. 

However, inflammation can also pose potential dangers. For example, patients with systemic inflammatory response syndrome may excessively secrete granzymes and perforins, leading to severe immune-related diseases like asthma and further resulting in multiple organ failure and disseminated intravascular coagulation. For instance, in the lungs, excessive sputum production can block the airways, leading to patient suffocation [15]; in the brain, inflammation can lead to deterioration of the central nervous system, causing irreversible cognitive impairment [16]; and inflammation in the heart can severely weaken blood pumping capacity, leading to myocardial function deterioration [17]. 

Following the elimination of infection factors, the macrophages transit from the cytotoxic pro-inflammatory phase to the anti-inflammatory, restorative phase, primarily through the recruitment of macrophages together with a small number of tissue-resident macrophages [18,19]. In earlier studies, it was assumed that neutrophils self-destruct or undergo apoptosis, leading to a decrease in inflammation. However, this belief has now been corrected. It is understood that the repairing macrophage actively eliminates the over-proliferated neutrophils during the pro-inflammatory phase. During the inflammatory process, the apoptosis of neutrophils triggers the recognition and clearance of dead neutrophils by macrophages, which are necessary to reduce inflammation [13]. Since the occurrence and healing of pneumonia through the biological mechanism of macrophages’ phenotypic transformation have been well investigated, we use pneumonia as a classic example to comprehensively discuss the correlation between macrophage transformations and the progress of inflammation.

### 2.1. The Balance between Pro-Inflammation and Anti-Inflammation in Pneumonia

Alveolar macrophages (AMs) play a critical role in protecting the lungs against pathogens and harmful particles in the air and help maintain the homeostasis of the alveolar environment (Figure 1). Many typical recognition receptors, such as the classical Toll-like receptor, bind through the NF-Kappa B factor, which is an essential biological mechanism for the self-protection of epithelial cells [20,21]. In addition to the release of inflammatory factors, proteins secreted by alveolar epithelium (type II) can help maintain alveolar surface tension to inhibit microbial reproduction [22]. Alveolar epithelium (Type II) functions as a cell reservoir that can replenish the damaged type I alveolar epithelium. When a pathogen invades the body [23], macrophages are rapidly polarized into pro-inflammatory cells, which can be cytotoxic to both pathogens and normal tissues. On the other hand, repair cells, such as fibroblast and Regulatory cells (Treg), quickly respond and reach the site to prevent or alleviate the inflammatory reaction and avoid excessive harm to the body. Thus, immune homeostasis is achieved through a balance between immune response and defense. During pneumonia, the immune imbalance is a primary pathogenic factor, with the severity of the disease being directly proportional to the degree of immune imbalance. Immune escape resulting from macrophages’ inability to effectively present antigens can easily lead to the spread of inflammation. In vivo experiments conducted on Staphylococcus aureus and Streptococcus pneumoniae have shown that ordinary pneumonia can quickly progress to acute lung injury (ALI) if immune defense ability is poor. ALI may further lead to the development of acute respiratory distress syndrome and sepsis [24].

Numerous chemokines, including C-C Motif Chemokine Ligand 2 (CCL2) and CCL10, have been discovered in the bronchoalveolar fluid of COVID-19 patients. These chemokines selectively recruit C-C Motif Chemokine Receptor 2(CCR2^+^) monocytes, which leads to a significant amount of aberrant macrophage aggregation [25]. The accumulation of a significant amount of inflammatory cells, persistent damage in epithelial cells of the lung, increased alveolar permeability, and diffused interstitial inflammation in the lung organ have been verified to be the typical pathological features of viral pneumonia induced by Severe Acute Respiratory Syndrom (SRAS)-COVID-2. Thus, the macrophages have to develop a pro-inflammatory phenotype and generate many inflammatory mediators, leading to a cytokine storm. Unfortunately, the morbid pyroptosis of macrophages results in the increased necrosis of the surrounding tissue/organ, which finally leads to persistent inflammation of multiple organs, including the nephron-urinary system, testicles-reproductive system and brain-nervous system, etc. 

On the other hand, persistent inflammation will lead to multiple organ failures and even mortality. According to the report by Dong Wu et al., the treatment of apoptotic gene caspase-1 inhibitors could dramatically decrease the production of IL-18, TNF-α, and other inflammatory cytokines in lipopolysaccharide-induced ALI mice [26]. In addition to caspase-1, the studies on the classical signal pathways about the differentiation of pro-inflammatory macrophage, for example, Toll-like receptor 4 (TLR4), NF-Kappa B, and p38-MAPK, have demonstrated that excessive expression of pro-inflammatory macrophages is closely linked to the cell pyroptosis [27,28,29,30]. In order to accurately treat ALI, the inhibitors could be employed to block the macrophages converting from resting state to pro-inflammatory types. It has to evolve into a novel immunotherapy strategy for pneumonia caused by serious inflammation [31].

Figure 1 displays diagrammatically the work of the immune system in pneumonia. NF-κB factor pathway will be activated when the alveolar epithelium (type I/II) gets injured because of a bacterial infection or virus invasion. As a tissue-resident, the alveolar macrophage is activated and secretes some pro-inflammatory factors like IL-17 or IL-22 [32]. These cytokines play roles through a cascade of reactions with the aim of phagocytosing and eliminating foreign pathogens. In this process, the B1a cell will secrete IgM to participate in mucosal immunity. Excessively strong or long-term inflammatory reactions will cause the abnormal activation of the mononuclear phagocyte system. Thereby, the excessive pro-inflammatory macrophages and neutrophils are recruited and go through the barrier of vascular endothelial cells [33]. These immune cells enter the alveoli and continue to damage normal tissues. The primary harm is caused by the cytokine storm and extensive fibrosis, which greatly deteriorate the normal function of the lung [34].

### 2.2. The Role of Macrophage Phenotypes in Response to Inflammation

The mononuclear phagocyte system consists of monocytes, macrophages and dendritic cells, among which macrophages are the professional phagocytes and non-specific antigen-presenting cells (APCs) capable of uptaking various pathogens that invade the body’s immune system. Additionally, they are secretory cells that participate in the inflammatory response and regulate the inflammatory process. Various phenotypes of macrophages have been identified to play distinct roles in different pathological processes to maintain homeostasis of the internal body environment. Under normal conditions, most macrophages travel around connective tissue, while a small number of macrophages settle in specific tissues and play an important immune role. These cells are called tissue-resident macrophages (TRMs). Examples include Kupffer cells in the liver, microglia in the central nervous system, osteoclasts in bone tissue, and others. Resting macrophages typically secrete very little cytokine and are, therefore, unable to effectively eliminate pathogens. Previous studies have identified two primary macrophage phenotypes, M1 and M2, which exist throughout the entire mononuclear phagocytic system. M1-like macrophages constantly express pro-inflammatory factors due to the overlapping gene expression induced by multiple cytokines and signal pathways. However, this paradigm can be imprecise in describing macrophage status and may interfere with disease diagnosis. In a recent study, 29 types of macrophages that differ from the classic M1/M2 types of macrophages were identified [7]. Following this result, some researchers have called for more rigorous descriptions of macrophage phenotypes to replace the previous M1/M2 dichotomy.

When the body is exposed to external threats, a range of receptors, including pattern recognition receptors (PPR), modulated receptors, chemotactic receptors, activated receptors, and antigen-presenting membrane receptors located at the cell membrane, are stimulated to participate in regulating macrophage phenotypes. Due to the plasticity of macrophages in response to fluctuations in the microenvironment, their roles in maintaining immunological homeostasis can be easily misunderstood. Therefore, we used the widely present inflammatory response in the body as an example to describe the functions of macrophages through the signal pathways activated by receptors on the cell membrane and the downstream cytokines in this article to clarify their important roles in the regulation of inflammation. 

The signal pathways related to macrophage polarization and the correlations among these pathways are presented in Figure 2. In this figure, we illustrate the differentiation and maturation of macrophages. At first, monoblasts are derived from multipotential stem cells in the bone marrow. They can differentiate into several subtypes after being stimulated by various molecules [35,36]. For example, monoblasts differentiate into osteoblasts and osteoclasts with the stimulation of macrophage colony-stimulating factor (M-CSF) and TNF Superfamily Member 11 (TNFSF11) or RANKL, which plays a vital role in the development and maturation of bone [37]. The dynamic balance between these subtypes is pivotal in regulating osteoblast growth, bone development, and remodeling. Once pro-monocytes enter blood vessels, monocytes can be divided into LY6C+, LY6C-, CD163+/CD206+, and CD80+/CD86+ cell lines according to the human leukocyte differentiating antigen in the cell membrane. Monocytes LY6C+ differentiate into dendritic cells under the action of IL-4, GM-CSF and M-CSF, which can resist external infection and present pathogen surface antigens, thereby activating cellular immunity [38]. LY6C-, CD163+/CD206+, and CD80+/CD86+ monocytes can develop into different cell lines mediated by different cytokines like interleukins and CCLs, and play unique roles in some specific diseases. For instance, IL-4 and IL-10-induced M2a macrophages can activate fibroblast growth, inhibit inflammation and promote the repair of damaged tissues. LPS, IFN-γ, IL-1β, TNF-α, and other pro-inflammatory factors induced by M1 macrophages can enhance the immune defense against the invaded pathogenic microorganisms. However, the abnormal release of these cytokines may unexpectedly lead to the dysfunction of macrophages, resulting in the occurrence of diseases. For instance, a large concentration of M1 cells can produce inflammatory storms and may even lead to shock in patients. Tumor-associated macrophages may induce the differentiation of Treg cells and cause tumor metastasis. Therefore, a normal and appropriate macrophage population is vital for body health.

#### 2.2.1. Pro-Inflammatory and Cytotoxic Macrophages

Classic macrophages, often referred to as M1-type macrophages, are pro-inflammatory macrophages that can activate classic pro-inflammatory pathways and secrete inflammatory mediators. They have significant impacts on cell toxicity, necrosis, and apoptosis. While the inflammatory response triggered by M1-type macrophages can eliminate pathogens that infiltrate the body, these pro-inflammatory macrophages may also uncontrollably differentiate and cause serious injury to the body. This phenomenon is exemplified by the illness caused by the SARS-CoV-2 virus, in which a number of macrophages gather in the lung and produce excess cytokines, leading to mass cell apoptosis and ultimately resulting in patient death. 

Classical macrophage polarization occurs when cells are stimulated by one or more of the following factors: (1) IFN-γ secreted primarily by Th1 cells, cytotoxic T cells and Nature-killer cells; (2) Lipopolysaccharide (LPS), the main component of the outer membrane of Gram-negative bacteria; and (3) Granulocyte-macrophage colony-stimulating factor (GM-CSF), which induces the differentiation of progenitor cells into pro-inflammatory macrophages, and effectively identify and kill tumor or/and virus-infected cells. Pro-inflammatory macrophages can be identified by their secretions of endogenous thermogenic factors such as IL-1β, TNF-α, IL-6, inducible nitric oxide synthase (iNOS), and inflammatory chemokines IL-12 and IL-18. Phenotypically, these macrophages express high levels of major histocompatibility complex class II (MHC-II), CD68 markers, and co-stimulatory molecules CD80 and CD86. Recent studies have also shown that pro-inflammatory macrophages mediate the expression of intracellular cytokine transduction inhibitors like Suppressor of Cytokine Signaling 3 (SOCS3), and activate iNOS to produce nitric oxide (NO) [39,40]. Therefore, an excessive number of pro-inflammatory macrophages will disrupt the body’s homeostasis, leading to cytotoxicity, apoptosis, or even pyroptosis.

#### 2.2.2. Signal Pathways Closely Associated with Pro-Inflammatory Macrophages

Biologically, there are several signal pathways related to the pro-inflammatory response of macrophages, some of which are summarized in the following sections.

##### NF-Kappa B Signal Pathway

NF-kappa B (NF-kB) is a multipotent transcription factor that exists in almost all cell types and mediates intracellular signal transduction as both an intermediate factor and a terminal effector. This signal factor is distributed in a variety of subcellular compartments and can be activated by many cytokines related to inflammation, innate and adaptive immunity, cell migration, proliferation, apoptosis, and tumorigenesis. The classical NF-Kappa B signal pathways, including RELA/P65, RELB, NF-Kappa B1/P50, NF-Kappa B2/P52, and C-REL [41,42,43], are displayed in Figure 3A. The abundance of NF-kB/P105, comprising NF-kB p65, and NF-kB p50, is the highest [44,45]. In the cytoplasm of unstimulated cells, the NF-kB complex is inhibited by binding to the NF-kB inhibitor (I-Kappa-B). In the canonical pathway, various cytokine receptors such as the tumor necrosis factor receptor (TNFR) [43] and interleukin receptor (IL-1R) [46] may activate the NF-kB signal pathway to alter the behaviors of cells and organelles and generate a special signal presentation. After receiving this presentation, I-Kappa-B is phosphorylated by I-Kappa-B kinase, which activates different downstream effectors, such as TNF-α, IL-1β, MIP-1β and MIP-2, through cascade amplification to participate in the pro-inflammatory process [47].

##### Jak/Stat and IL-6 Receptor Family Signal Pathways

Interleukin 6 (IL-6), a downstream factor of the IL-6 receptor family pathway, plays a role in the early phase of inflammation and stimulates B cell maturation [48]. Additionally, IL-6 is an endogenous pyrogen that can induce fever in patients with autoimmune diseases or infections. It is primarily produced at the site of inflammation and secreted into the serum, inducing a transcriptional inflammatory response via the IL6Rα receptor. IL-6 is genetically related to various inflammatory diseases like diabetes and systemic juvenile rheumatoid arthritis. Elevated levels of this protein have been found in virus-infected patients, particularly those infected by SARS-CoV-2 [49]. 

IL-6 has various biological functions in immunity, tissue regeneration, and metabolism. During an infection, this complex binds to the IL-6 receptor (IL6R) and the signal subunit, Interleukin 6 Cytokine Family Signal Transducer (IL-6ST)/GP130, triggering the intracellular STAT3 signal pathway (Figure 3B). The interaction between IL-6ST and membrane-bound IL6R stimulates “classical signal transduction,” while the combination of IL-6 and soluble IL6R with IL6ST stimulates “trans-signal transduction” [50]. Additionally, “cluster signaling” occurs when the membrane-bound IL6R complex on the transmitter cells activates the IL-6ST receptor on adjacent receiver cells. 

IL-6 is also an effective inducer of the acute inflammatory response. During an infection, IL-6 production contributes to host defense, but an excessive synthesis of IL-6 is involved in disease pathology. In the innate immune response, myeloid cells such as macrophages and dendritic cells recognize pathogens through Toll-like receptors (TLRs) at the site of infection or injury. During an infection, IL-6 induces the synthesis of liver proteins, including C-reactive protein (CRP), complement C3, fibrinogen, thrombogenic proteins, serum amyloid A, and fibromodulin, during the acute phase, which also inhibits albumin production [32]. In addition, endothelial cells are activated to produce IL-6, IL-8, monocyte chemoattractant protein-1 (MCP-1), intercellular adhesion molecule-1 (ICAM-1), and C5a receptor, while endothelial cadherin is dissociated. 

However, excessive or prolonged secretion of IL-6 can lead to the development of various diseases. For instance, in the early stage of hepatocyte injury, Kupfer cells secrete TNFα, inducing a large amount of IL-6 via autocrine signaling. This excessive amount of IL-6 subsequently stimulates liver cells to proliferate. Therefore, IL-6 tests can help identify the severity of a disease but cannot determine its etiology. Due to its high sensitivity to infection and fever, IL-6 can be used as an early indicator for controlling inflammation. When the treatment is effective, it can also serve as an excellent prognostic indicator compared to others, such as IL-1β. 

##### Toll-Like Receptor Signal Pathway

Toll-like receptors (TLRs) play an irreplaceable role in the innate immune response by recognizing different pathogen-associated molecular patterns (PAMPs). They serve as the first line of defense against pathogen invasion and are essential for inhibiting inflammation, regulating immune cells, and promoting cell survival and proliferation during inflammation. The cytoplasmic Toll/IL-1 receptor (TIR) domain, which interacts with the adaptor protein MyD88, is responsible for the activation of the TLR signal transduction pathway. For instance, the LPS found in the cell walls of Gram-negative bacteria can trigger this signaling system. CD14 is the most important LPS receptor on the cytoplasmic membrane of immune cells. Once activated by a ligand, MyD88 attracts IL-1 receptor-associated kinase-4 (IRAK-4) to TLRs through their interaction via the two molecular death domains [51]. Upon activation by phosphorylation, IRAK-1 interacts with TRAF6, which in turn activates the IKK complex, leading to the activation of MAP kinases (JNK, P38 MAPK) and NF-kappa B [52] (Figure 3C). 

Macrophages have a wide range of Toll-like receptors on the cytoplasmic membrane that allows them to express various proteins, including cytokines and chemokines, in response to pathogenic stimuli. During Gram-negative bacterial infections, the early production of proteins like NFKBIZ and ATF3 has been observed [53]. Additionally, it has been shown that TLRs clusters, such as NFKBID and BCL6 [54], participate in the tardive immune response and recruit pro-inflammatory macrophages. The differentiation of macrophage subtypes has been correlated with other TLR-inducible genes, such as JMJD3 and TRIB1 [55,56]. JMJD3 plays a significant part in controlling the M2 type of anti-helminth host responses. Additionally, TRIB1 can affect the actions of tissue-dwelling macrophages to keep adipose stem cells stable during tissue regeneration.

#### 2.2.3. Anti-Inflammatory and Restorative Macrophage

Anti-inflammatory macrophage, also known as alternatively activated macrophage or M2 macrophage, has been found to play a regulatory role in tissue remodeling, angiogenesis, allergic diseases, and parasitic infections. Studies on anti-inflammatory macrophages have shown that certain stimuli, such as CSF-1, IL-4, IL-10, TGF-β, and IL-13 cytokines, can promote the polarization of macrophages during the later stages of the inflammatory response. Additionally, recent studies have found that tumor-infiltrating macrophages, similar to anti-inflammatory macrophages, are associated with tumor metastasis. 

Phenotypically, this M2 subpopulation is mainly expressed on the macrophage mannose receptor (MMR) or CD206. However, Jaguin et al. reported no significant difference in CD206 expression level between M1 and M2 cells, suggesting that the up-regulation of CD206R membrane glycoprotein could be responsible for the specific expression of M2 macrophages [57]. CD163, a marker of M2 macrophages, plays a role in differentiating macrophages from the M1 to M2 phenotype [58,59,60]. However, differentiating between macrophage types solely by cellular markers on the membrane is insufficient. Thus, a comprehensive analysis of the anti-inflammatory cytokines secreted by macrophages is essential for diagnosing and treating diseases.

#### 2.2.4. Signal Pathways That Are Tightly Linked to Anti-Inflammatory Macrophages

##### TGF-β Signal Pathway

TGF-β is a multifunctional protein family that regulates the growth and differentiation of all types of cells and participates in various processes, such as embryonic development, immune regulation, and response to neurodegenerative diseases. The downregulation of TGF-β can easily lead to various diseases, including fibrosis, chronic inflammation and malignant tumors [61]. Recent studies have shown that both regulatory T cells (Treg) and B cells (Breg) can produce TGF-β through the endocrine pathway, which can promote the transformation of the macrophages from pro-inflammation into anti-inflammation [62,63]. 

TGF-β1, one member of the TGF-β family, is released from Latency associated peptide (LAP) by the integrin and then binds to the receptors, TGFβR1 and TGFβR2, to activate the downstream molecules. Figure 4A demonstrates that TGF-β1 can activate the protein BMP2 upon binding with Smad2/3, resulting in the chemotaxis, proliferation, and differentiation of osteoblasts and ultimately achieving bone tissue remodeling. Additionally, TGF-β can promote the cell lineage differentiation of Treg or Th17 cells to regulate the immune response in a concentration-dependent manner. Under high TGF-β1 concentration, FOXP-mediated RORC inhibition and IL-17 down-regulation promote Treg cell differentiation. At low concentrations, TGF-β1 stimulates macrophages to synergistically secrete IL-6 and IL-12, leading to the expression of CCL2 and MIP-1. This subsequently stimulates the mononuclear phagocyte system to secrete IL-17 and IL-23 receptors, promoting Th17 differentiation and strengthening inflammatory defense. Additionally, TGF-β1 synergizes with other cytokines to play important roles in epithelial-mesenchymal transition (EMT) and cell migration in various cell types.

##### IL-10/mTOR Signal Pathway

IL-10 is a major cytokine that plays a crucial role in modulating the immune response. It acts on T, B, and dendritic cells, exerting a potent anti-inflammatory effect that helps limit the tissue damage caused by inflammatory reactions. Studies have revealed that IL-10 binds to a heterotetrameric receptor composed of IL-10Ra and IL-10Rb, leading to the phosphorylation of STAT3 through JAK1 and STAT2-mediated signaling pathways (Figure 4B). Once phosphorylated, STAT3 is translocated to the nucleus, where it binds to specific genes and promotes the transcription of anti-inflammatory mediators. The targeted cells, particularly macrophages, suppress the transcription and release of pro-inflammatory cytokines such as Peroxisome Proliferator-Activated Receptor-α, GM-CSF, G-CSF, IL-6, IL-1α, IL-1β, and TNF-α [64,65]. Furthermore, IL-10 interferes with the expression of MHC-Ⅱ and co-stimulatory molecules on the membrane, inhibiting their ability to activate T cells [66]. 

IL-10 can be reprogrammed via mTOR signal transduction to regulate nutrient metabolism pathways, thereby controlling the inflammatory responses of macrophages. Due to their ability to inhibit mTOR, mTOR inhibitors have been explored as potential immunosuppressive agents in organ transplantation, as well as their therapeutic potential against SARS-CoV-2 infection [67].

##### Arginine-Polyamines-Hypusine Pathway

Arginine (ARG) is an essential regulator of the immune response. In the immune microenvironment, neutrophils are released from the phagosome following cell apoptosis, leading to continuous consumption of arginine, which inhibits the proliferation and cytokine secretion of natural killer cells and T cells (Figure 4C). The impact of arginine on human monocytes, macrophages, and dendritic cells was not known until 2016 when Bossche et al. demonstrated that NO produced due to mitochondrial damage was closely associated with the transformation of macrophages from pro-inflammatory M1 type to M2 polarization [68]. Ji et al. also discovered that creatine uptake dysfunction in mice lacking the Slc6a8 gene could inhibit IFN-γ-mediated phosphorylation of STAT1. Moreover, when IL-4 was used to promote M2 differentiation, the expression of the ARG-1 gene was significantly upregulated [69]. Therefore, we concluded that the competition of pro-inflammatory iNOS and anti-inflammatory arginase for arginine substrate plays a role in regulating the dynamic balance of arginine in the whole body. The competition between pro-inflammatory iNOS and anti-inflammatory arginase for the arginine substrate plays a crucial role in regulating the dynamic balance of arginine in the body. It is noteworthy that eukaryotic translation initiation factor 5A (eIF5A) could suppress the activation of the oxidative phosphorylation-dependent macrophage alternatives. eIF5A participates in mitochondrial metabolism through eIF5A hypusination. The ability of polyamine biosynthesis to regulate macrophage activation has been revealed by Puleston et al. [70,71]. More thorough research is needed to confirm how it interacts with arginine in the process of macrophage polarization, as it is one of the arginine’s downstream metabolites.

All these signal pathways related to the phenotypic transformation of M1/M2 macrophages and their biological functions in wound healing or disease therapy are summarized in Table 1.

#### 2.2.5. Crosstalk between Various Signal Pathways

Due to their diversity and good phenotypic plasticity, macrophages are highly susceptible to phenotypic transformations under various stimulations in the immune microenvironment. During the early stages of inflammation, macrophages in blood and tissues recognize antigenic substances from the invading pathogens, such as LPS and invasive enzymes, and activate TLRs on their cell membrane, causing polarization into a pro-inflammatory phenotype. The activation of NF-Kappa B and JAK signal pathways subsequently promotes the phosphorylation of transcription factors, which translocate to the nucleus and bind to pro-inflammatory genes, leading to the high-level expression of inflammatory cytokines such as IL-1, TNF-α, and IL-6. Meanwhile, due to the inactivation of inhibitory factors such as IKappa B, the phosphorylated inflammatory transcription mediators are not promptly cleared, resulting in the continuous inactivation of STAT3 or mTOR. Consequently, many downstream genes involved in the transition of macrophages to an anti-inflammatory phenotype, such as the above-mentioned JMJD3, TRIB1, and Slc6a8, cannot be normally expressed. This inability to translate important negative regulatory molecules of the immune response during the later stages of the inflammatory response is closely associated with the chronic development of inflammation.

Pro-inflammatory macrophages are associated with various proteins, such as MAPK products, P38, Ras protein, and other signaling molecules that promote the expression of MHC-Ⅱ on the membrane, interfere with the inflammatory response, and stimulate immune cells to release numerous inflammatory factors to combat pathogenic microorganism. However, excessive aggregation of inflammatory factors can cause irreversible and persistent damage to normal cells and disrupt the normal physiological functions of tissues and even organs through cascade amplification, ultimately resulting in immune system instability. Therefore, a drug intervention is necessary to regulate the inflammatory cytokine storm. For example, in the treatment of acute respiratory distress syndrome (ARDS) caused by the SARS virus, the use of antioxidants and pathway inhibitors has improved the pulmonary condition of patients, demonstrating the feasibility of using novel agents to regulate the inflammatory process.

Once the pathogens are eliminated, the body enters the restorative phase. On the one hand, IKappa B rebinds to NF-Kappa B and inhibits its phosphorylation, preventing NF-Kappa B from entering the nucleus. Meanwhile, STAT3 is successfully translocated to the nucleus, promoting the release of repair factors such as IL-10, IL-13, and IL-35. Additionally, Treg and other regulatory immune cells further promote the release of TGF-β1, express FasL, CD1b, and other membrane regulatory molecules to induce the apoptosis of neighboring neutrophils and negative inhibitory factors, CTLA-4, and PD-1, to initiate inhibitory signals. Together, these mechanisms inhibit the immune response [72]. On the other hand, TGF-β1 induces positive feedback through the phosphorylation of Smad2 and Smad3 of ActivinR I and ActivinR II, leading to fibroblast activation, self-renewal of stem cells, and ultimately, the repair of damaged tissues and restoration of organ function [73,74]. These cross-linked pathways are schematically diagrammed in Figure 5.

The distinction between M1 and M2 macrophages in six tissue healing processes is expounded in Figure 5. When tissues are damaged by external factors, a large number of pathogens would infiltrate. The resident macrophages on the tissue serve to signal transduction during this period, activating M1 macrophages and secreting numerous inflammatory cytokines. After all of the infections have been eliminated, M1 macrophages progressively convert into M2 macrophages, which produce substances to reduce inflammation, attract Breg cells and Treg cells, and inhibit the immune response.

## 3. Recent Research on Inflammation Control

### 3.1. Why Do We Interfere with Inflammation Response

Acute inflammation is a local host response to pathogen attacks and serves as a safeguard for the body. However, if inflammation is uncontrollable, the immune system can trigger universal systemic inflammation. In clinical settings, some of the most difficult-to-manage ailments include pneumonia, asthma, cardiovascular disease, acute progressive glomerulonephritis, and acute suppurative inflammation of the nervous system. Treating these disorders requires addressing inflammation clearly and taking proper measures immediately to inhibit inflammatory deterioration, preventing excessive or persistent inflammation in the host. Long-term clinical experiences have revealed that the use of traditional antibiotics might result in substantial, irreversible damage to the host. Indeed, all chronic refractory diseases or malignant tumors are generated from excessive or persistent inflammation in the body. This predicament motivates the research of novel and functional therapies other than antibiotics. Since the outbreak of SARS-CoV-2 in 2019, patients infected with the COVID-19 virus have typically presented with acute respiratory distress syndrome (ARDS) and severe storms of inflammatory factors, which caused numerous deaths worldwide. Under these circumstances, the most urgent strategy is to reduce excessive inflammatory responses to protect the lung from extreme injury.

### 3.2. Adverse Effects of Long-Term Application of Anti-Inflammatory Drugs

Since the discovery of penicillin and other antibiotics, many broad-spectrum antibiotics have been used in the early stages of inflammation, including pneumonia, meningitis, endocarditis, diphtheria, anthrax, and other diseases. However, in the analysis of clinical data, Klompas et al. discovered that physicians often prescribe antibiotics before a positive microbial culture, even in cases of early inflammation [75,76]. In China, 78% of young children who visited clinics for common colds were reportedly given antibiotic prescriptions [77]. Unexpectedly, the side effects of antibiotics can be lethal. A vast amount of clinical evidence indicates that the empirical use of broad-spectrum medications may be directly linked to increased mortality. It is essential to emphasize that not all inflammatory reactions are caused by microbial contamination, and those caused by viruses are largely ineffective with antibiotics. In the early phases of inflammation, if the diagnosis of inflammation, such as pneumonia, is uncertain, less harmful therapies should be employed instead of antibiotics.

Other kinds of anti-inflammatory drugs, such as glucocorticoids (GC) and non-steroidal drugs like aspirin, are also widely used in clinical treatments [78]. Short-term usage of glucocorticoids and aspirin effectively inhibits inflammation, but long-term usage may lead to irreversible kidney injury [79]. Many side effects of glucocorticoids are presented in major endocrine organs, causing kidney dysfunction [80] and further affecting the skin and cardiovascular and gastrointestinal systems [81]. Up to 90% of patients who use glucocorticoids for more than 60 days may experience water-sodium retention and hypopotassemia [82]. Clinical studies have confirmed that long-term usage of aspirin can easily lead to gastrointestinal ulcers [83]. Additionally, researchers have found that aspirin administration may aggravate the risk of asthma, which is not helpful in treating pneumonia [84]. Therefore, it is urgent to find effective inflammation-defense measures instead of relying solely on traditional medicine.

### 3.3. dECM-Related Material Is a Novel Therapy to Moderate the Inflammation

The extracellular matrix (ECM) comprises various structural and functional components, including glycoproteins, elastin, collagen, and proteoglycans. These components work together to create a dynamic extracellular microenvironment involved in intracellular signaling, cell migration, and differentiation. ECM from mammalian sources has excellent histocompatibility, allowing for preserving natural tissue structure and cytokines. It is achieved by removing substances that can cause immune rejection, such as MHC molecules and DNA, and by maintaining a ratio of various substances like collagen, hyaluronic acid, and elastin. The proportions of collagen, hyaluronic acid, and elastin in the ECM are similar to those found under normal physiological conditions, as confirmed by in vivo and in vitro experiments over the past several decades, more details can be found in Table 2.

Decellularized materials from different tissues have their specific tissue components. In addition to the aforementioned collagen, elastin and macromolecular polysaccharides, many kinds of specific cytokines can connect with the specific proteins in the main components of most tissues, such as fibroblast growth factor (FGF), transforming growth factor (TGF), vascular endothelial growth factor (VEGF), bone morphogenetic protein (BMP-2), endothelial cell growth factor (EGF), etc. It is also reported that many growth factors, such as TGF-β, FGF-2, and VEGF, exist in the small intestinal submucosa (SIS) [85,86], bone morphogenetic protein (BMP-2), and endothelial cell growth factor (EGF), etc. In addition to glycosaminoglycan (GAG) and collagen, the acellular pig skin also contains immune regulatory factors such as CD26, FGF-1, FGF-2 and IGFBP-9 [87]. 

Several studies have confirmed that scaffolds containing natural extracellular matrix (ECM) have the potential to promote tissue regeneration and regulate innate and adaptive immune responses, reducing inflammation and scar formation. There are also works of literature indicating that decellularized ECMs (dECMs) can regulate the balance between pro-inflammatory and anti-inflammatory macrophages, thereby shortening the inflammatory process in local tissues. References [88,89,90] report that dECMs can regulate the proportion of pro-inflammatory macrophages and anti-inflammatory macrophages to shorten the inflammatory process in local tissues. T cells are also regulated by dECMs to differentiate into Th2 or/and T^reg^ cells, which play a synergistic stimulating role and jointly regulate immune defense with anti-inflammatory macrophages [91]. In addition, it is worth noting that matrix binding nanovesicles (MBV) are found to be an important part of dECMs because the bioactive substances contained in MBV can accelerate the growth and proliferation of cells, similar to the extracellular vesicles (EVs) [92]. In regard to managing inflammation, both dECMs and EVs have demonstrated the ability to regulate the immune system. Specifically, they were found to influence the polarization and transformation of macrophages and help to moderate inflammatory responses, which is particularly important as it offers an alternative to the potential adverse effects associated with the long-term use of traditional drugs.

Our research group has extensively studied the effects of decellularized human amniotic membranes and pig gastric submucosa. The immunohistochemistry results demonstrated that the increased secretion of VEGF and α-SMA, along with the decreased secretion of TGF-β1 during the early stages of wound healing, facilitate the recruitment of anti-inflammatory cells and clear the early infection. Thus, animals treated with dECM showed a significantly faster repair rate than non-treated animals [93,94,95]. At the RNA level, dECMs were shown to promote the mRNA expression of leucine-rich and immunoglobulin-like structural domain 1 (LRIG1) and inhibit the mRNA transcription of TGF-β1 [96]. In the other study, dECMs were able to inhibit the overexpression of the NF-κB pathway (not published). In addition, dECMs combined with hyaluronic acid promote the conversion of macrophages to M2 phenotype and secrete immunosuppressive cytokines IL-10 and ARG, thus accelerating tissue repair. Collectively, these findings provide compelling evidence of the significant potential for dECMs in clinical applications.

**Table 2 ijms-24-08358-t002:** ECM-related material promoting disease therapy or tissue regeneration.

Materials	Seeding Cells	Diseases	Function	References
Decellularized lung organ	Lung epithelial and endothelial cells	Chronic obstructive pulmonary	Establishing a model of the lung’s physiological microenvironment to carry out gas exchange.	[97]
Decellularized liver matrix	Hepatocytes	Acute and chronic liver damage	Supporting hepatocyte survival and function like albumin secretion, urea production, and cytochrome P450 expression.	[98]
Decellularized kidney	Renal endothelial cells	Chronic kidney disease, end-stage renal disease	Clearing metabolites, reabsorbing electrolytes, and generating concentrated urine.	[99,100]
Perfusion-decellularized whole-heart scaffolds	Myocardial cells induced by iPSC	Heart failure, myocardial infarction	Regulating angiogenic growth factors and guiding anisotropic microvascular growth and development towards maintaining heart homeostasis and remodeling.	[101,102]
Decellularized pig skin combined with gelatin/hyaluronic acid	Fibroblasts	Diabetic foot ulcers, large area skin trauma	Promoting granulation tissue formation, epithelial regeneration and pro-angiogenesis activity. Reducing scar formation by shortening the inflammatory stage.	[95,103]
Decellularized cow and human cadaveric bone	Bone marrow stromal cells	Bone fracture, osteoarthritis	Promoting chondrocytes differentiation, maturation and osteogenics to improve repairing long bone defects.	[104,105]

## 4. Conclusions and Prospective

Recent studies on the relationship between immune response and inflammatory processes have shifted towards multi-protein interaction networks that include protein crosstalk and cascade amplification of associative signal pathways, in which macrophages were reported to play crucial roles. The production and release of some signal factors have been connected to the transformation of cells’ phenotypes. In addition, the composition of these factors can operate as important, detectable molecules in the inflammatory process and aid in the diagnosis of diseases. The main mechanisms involved in the activation of various kinds of macrophages were outlined in this review. It is important to note that different cell states can exhibit diverse responses to the same cytokine due to the complexity of signal pathways. While emerging monitoring techniques such as proteomics can monitor a large number of cytokines, they often lack accuracy. Therefore, researchers must integrate multiple cytokines and consider various situations to achieve a precise diagnosis and treatment for patients.

The dysregulation of the immune system in response to inflammation can have serious consequences. Excessive immune defenses in the lungs can create a cytokine storm, leading to necrosis, apoptosis, or severe organ failure. Conversely, an excessively low immune capacity of the body can result in the breeding of pathogenic microorganisms, leading to infection and even sepsis. Therefore, the intervention of anti-inflammatory substances or pro-inflammatory cytokines is necessary to regulate the normal response of the immune system. For example, the combination of extracellular matrix and cytokines can be directed to regulate tissue structure and functional remodeling. However, this novel treatment is still in the experimental stage; the type of cytokine and the amount of extracellular matrix that should be applied to different stages of the disease are still being investigated. Presently, the potential adverse effects of this experimental therapy remain undetermined. Thus, to increase the safety of using dECMs and cytokines, it is necessary to have a comprehensive understanding of the mechanisms and behaviors of macrophages in multiple organ systems.

## Figures and Tables

**Figure 1 ijms-24-08358-f001:**
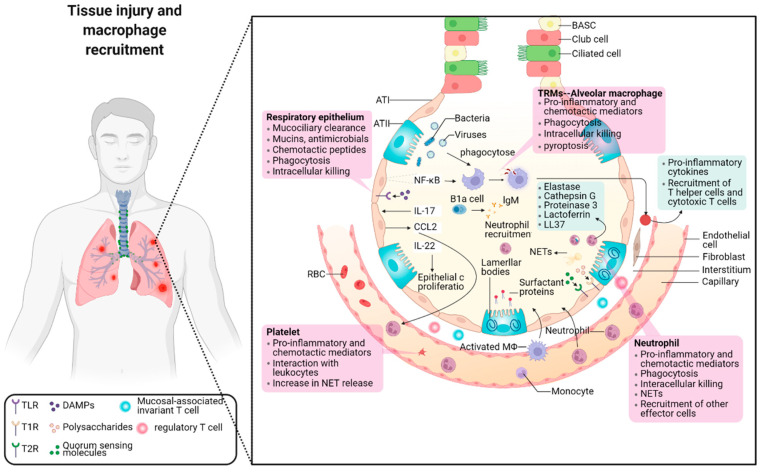
The schematic diagram of the immune system in pneumonia. It exhibits the procession of lung deterioration and macrophage recruitment. As pneumonia is a complex inflammatory response, the local inflammation will have a detrimental effect on the entire body if the immune system is not able to handle it properly.

**Figure 2 ijms-24-08358-f002:**
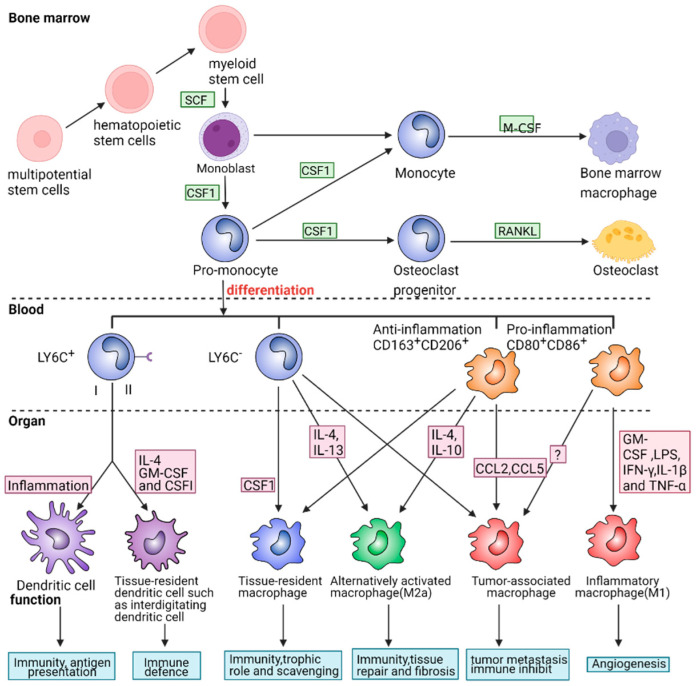
Differentiation of macrophage phenotypes. In order to maintain body health, the macrophage population must be normal and appropriate. As explained in this figure, maturation and differentiation of macrophages are closely associated with the function of the immune system.

**Figure 3 ijms-24-08358-f003:**
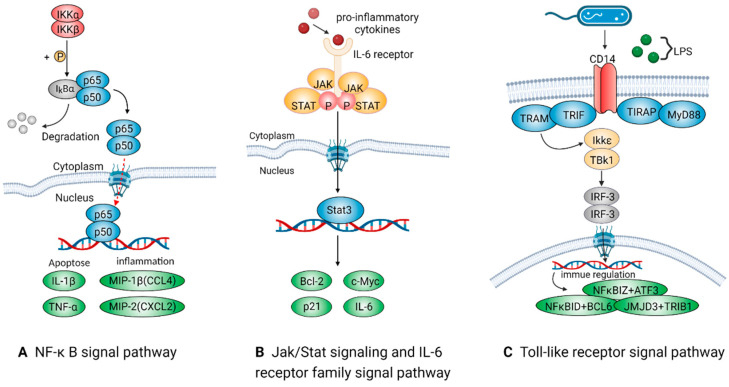
The pro-inflammatory signaling pathways activated by macrophages mainly include NF-Kappa B, Jak/Stat, IL-6 receptor family, and the Toll-like receptor signal pathway. The activation of these pathways has been strongly associated with the transformation of pro-inflammatory macrophages. The key protein molecules in these pathways are widely studied in the progression of inflammatory transformation.

**Figure 4 ijms-24-08358-f004:**
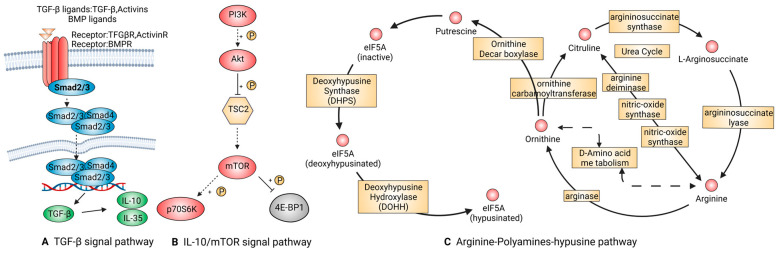
Signaling pathways that promote the shift of macrophages to an anti-inflammatory phenotype include but are not limited to TGF-β (**A**), IL-10/mTOR (**B**), and the Arginine-polyamines-hypusine pathway (**C**). These pathways are closely associated with T^reg^ and B^reg^ cells and thus are used to diagnose the inflammatory process and disease healing.

**Figure 5 ijms-24-08358-f005:**
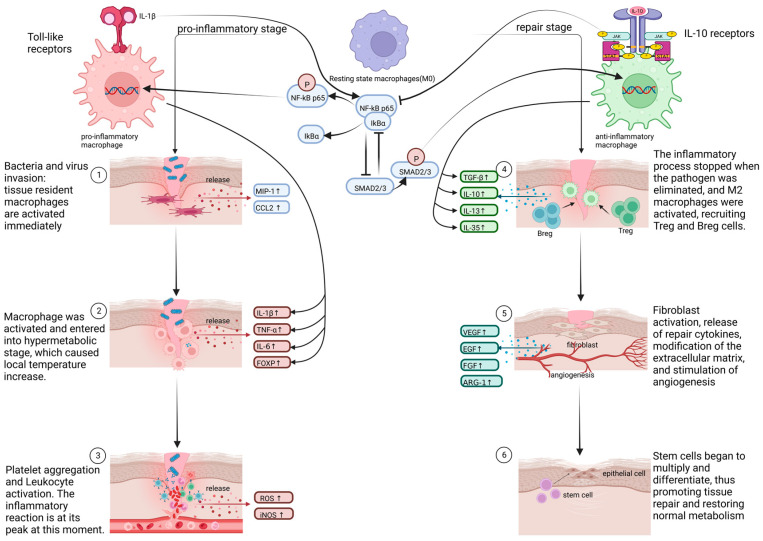
The function of macrophage phenotypic differentiation and the crosstalk of signaling pathways in wound repair. Inflammatory signals within the immune system, as well as metabolic imbalance, are the factors that support macrophage phenotypic alteration. Despite the fact that rapid macrophage recruitment and activation are capable of fending off the exterior pathogenic organisms, excessive macrophages will worsen tissue if pro-inflammatory macrophages are not quickly reined or changed into repair macrophages. Therefore, the recruitment and polarization of monocytes and macrophages must be strictly regulated during all kinds of tissue injury.

**Table 1 ijms-24-08358-t001:** Summary of the signal pathways related to M1/M2 macrophage and the biologic functions.

Phenotype of Macrophage	Signal Pathway	Factors	Functions	References
M1(pro-inflammatory)	NF-κB	NF-κB, TNF-α, IL-1β	Relating to inflammation, apoptosis and tumorigenesis. Activated by TNF receptor and IL receptor (IL-1R).	[41,42]
	Jak/Stat and IL-6 receptor family	STAT3, IL-6	Inducing fever in autoimmune diseases, acute inflammatory response and infections. IL-6ST/GP130 could trigger the intracellular STAT3 signal pathway, exacerbating inflammation.	[48,50]
	Toll-like receptor	TLR4, CCL2	LPS in Gram-negative bacterial walls could trigger this signaling system and take part in the tardive immune response or recruit pro-inflammatory macrophages.	[52,53]
M2(anti-inflammatory)	TGF-β	TGF-β1, BMP-2, Smad2/3	Regulating cell growth and differentiation and promoting transformation from pro-inflammatory into anti-inflammatory macrophages.	[55,56,57]
	IL-10/mTOR	IL-10	Taking actions on T, B, and dendritic cells in the immune system and exerting a powerful anti-inflammatory effect, limiting tissue destruction caused by inflammatory responses.	[59,60]
	Arginine-polyamines-hypusine	ARG, eIF5A	ARG is related to the transformation of macrophages from M1 to M2 polarization. eIF5A suppresses oxidative phosphorylation-dependent macrophage activation.	[66,67,68]

NF-κB, Nuclear Factor Kappa B; TNF-α, Tumor Necrosis Factor-alpha; IL-1β, Interleukin 1 Beta; STAT3, Signal Transducer and Activator of Transcription 3; IL-6, Interleukin 6; TLR4, Toll-like receptor 4; CCL2, C-C Motif Chemokine Ligand; TGF-β1, Transforming Growth Factor Beta 1; BMP-2, Bone Morphogenetic Protein 2; Smad2/3, SMAD Family Member 2/3; ARG, Arginase; eIF5A, Eukaryotic Translation Initiation Factor 5A.

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
