# Peer review of "Biologic Mechanisms of Macrophage Phenotypes Responding to Infection and the Novel Therapies to Moderate Inflammation"

_ijms, 2023, doi:10.3390/ijms24098358_

Round 1
Reviewer 1 Report
In this article, Ni et al. review the biological processes involved in macrophage interaction with neighboring cells and tissues in inflammatory diseases and the interaction of different signaling pathways controlling macrophage polarization. They also discuss the therapeutic approaches to regulate the inflammatory states of the macrophages. The article comprehensively discusses the major factors involved in the macrophage response to injury and infection. The diagrams are nice are descriptive. The authors need to address these minor comments before the acceptance of the article:
- Although the article covers important topics comprehensively, the major caveat of this study is the writing style and grammar. Because of the grammatical errors, including sentence structure, tenses, spelling, unnecessary spaces, etc., it takes time to understand what the authors want to convey. A thorough proofreading of the entire article by a native English speaker is warranted.
- Figure 4: Arginine also gives rise to the polyamines-hypusine pathway. In fact, the role of polyamines and hypusine has also been shown in macrophage polarization. The authors need to add a few lines on this pathway and add it to the figure as well.
- Figure 5: point 2 in the figure looks incomplete. Also, JAK and STAT3 are not clearly visible.
There are grammatical errors, including sentence structure, tenses, spelling, unnecessary spaces, etc. that need to be corrected throughout the article.
Author Response
Manuscript ID: ijms-2351781
Title: Biologic mechanisms of macrophage phenotypes responding to infection and the novel therapies to moderate inflammation
Dear Dr. Kamonlak Pongpanumaporn
Assistant Editor
IJMS
Thank you very much for giving us an opportunity to revise our manuscript. We appreciate the editor and reviewers very much for their constructive comments and suggestions on our manuscript (ijms-2351781).
We have studied all comments carefully and tried to fix all the problems mentioned. According to the reviewers’ detailed suggestions, we have made a careful revision on the manuscript. All revised portions are marked in red in the revised manuscript. The language of the manuscript was edited by Freescience Editorial Team.
Point-by-point response to the comments by Editor and reviewers
Reviewer 1#
Comments and Suggestions for Authors
In this article, Ni et al. review the biological processes involved in macrophage interaction with neighboring cells and tissues in inflammatory diseases and the interaction of different signaling pathways controlling macrophage polarization. They also discuss the therapeutic approaches to regulate the inflammatory states of the macrophages. The article comprehensively discusses the major factors involved in the macrophage response to injury and infection. The diagrams are nice are descriptive. The authors need to address these minor comments before the acceptance of the article:
Although the article covers important topics comprehensively, the major caveat of this study is the writing style and grammar. Because of the grammatical errors, including sentence structure, tenses, spelling, unnecessary spaces, etc., it takes time to understand what the authors want to convey. A thorough proofreading of the entire article by a native English speaker is warranted.
Figure 4: Arginine also gives rise to the polyamines-hypusine pathway. In fact, the role of polyamines and hypusine has also been shown in macrophage polarization. The authors need to add a few lines on this pathway and add it to the figure as well.
Res: Figure 4 has been revised. Arginine-polyamines-hypusine pathway was amended as (C) and the discussion was made in the section of "Arginine-polyamines-hypusine pathway".
Figure 5: point 2 in the figure looks incomplete. Also, JAK and STAT3 are not clearly visible.
Res: Figure 5 has been corrected. The figure legendary was also rewritten.
Comments on the Quality of English Language
There are grammatical errors, including sentence structure, tenses, spelling, unnecessary spaces, etc. that need to be corrected throughout the article.
Res: The language of the manuscript was edited by Freescience Editorial Team.

Reviewer 2 Report
On request of IJMS, I have revised the manuscript titled “Biologic mechanisms of macrophage phenotypes responding to infection and the novel therapies to moderate inflammation”, by Renhao Ni, et al.
The scope of this paper was to provide a review of the biological processes related to the macrophage’s interactions with surrounding cells and tissues during inflammatory diseases, and to clarify the signal pathways relevant to the phenotypic metamorphosis of macrophages. According to the authors, such information could facilitate the diagnosis of pneumonia and other disorders involving macrophages. A few novel therapeutic methods to regulate inflammation by controlling macrophage phenotypic transition so as to prevent the long-term negative effects of conventional medicine utilized in clinics have been also provided.
General comments
First, in my opinion, 8 authors are too much for a Review.
Anyway, the topic and the contents of the present manuscript could be interesting, but both the English language and the organization of the Review are poor and need improvement. Additionally, some minor issues need correction.
Following my opinion and suggestions.
The abstract is badly written and not clear. Please, reformulate and improve it.
Introduction and related references are poor. The background provided by the authors is very limited. More information is necessary to the readers to afford the topic.
As a review, the manuscript lacks completely Tables. As for the Figures, at least 5 Tables are necessary for a good work. So, authors are asked to include part of the information (the most suitable) in at least 5 Tables.
A good review work should contain up-dated references. Generally, over the 50% of references should be of the last five year. No reference of 2023 is present and more that 57% of references are older than the year 2018. Please, update the references.
Many Figure captions are excessively long (especially those of Figure 1, 2 and 5) and a reduction is necessary. Please, move the content of lines 132-142, 177-199, 306-309, 387-388 and 435-447 in the main text.
Minor
The numbers related to the affiliations in the list of authors should be superscripts.
Please, check all manuscript and correct when the space between words and references in the square brackets are missing, while remove spaces between numbers in the references.
Please, check all manuscript and specify all not specified abbreviations at their first mention.
Please, insert the correct information in Funding, Acknowledgements, and Conflict of interest sections.
Please, rewrite the reference list according to the instruction for authors of IJMS.
Sometimes, expecially in the abstract the concepts are not clearly expressed.
Author Response
Thank you very much for giving us an opportunity to revise our manuscript. We appreciate the editor and reviewers very much for their constructive comments and suggestions on our manuscript (ijms-2351781).
We have studied all comments carefully and tried to fix all the problems mentioned. According to the reviewers’ detailed suggestions, we have made a careful revision on the manuscript. All revised portions are marked in red in the revised manuscript. The language of the manuscript was edited by Freescience Editorial Team.
Point-by-point response to the comments by Editor and reviewers
Reviewer 2#
Comments and Suggestions for Authors
On request of IJMS, I have revised the manuscript titled “Biologic mechanisms of macrophage phenotypes responding to infection and the novel therapies to moderate inflammation”, by Renhao Ni, et al.
The scope of this paper was to provide a review of the biological processes related to the macrophage’s interactions with surrounding cells and tissues during inflammatory diseases, and to clarify the signal pathways relevant to the phenotypic metamorphosis of macrophages. According to the authors, such information could facilitate the diagnosis of pneumonia and other disorders involving macrophages. A few novel therapeutic methods to regulate inflammation by controlling macrophage phenotypic transition so as to prevent the long-term negative effects of conventional medicine utilized in clinics have been also provided.
General comments
First, in my opinion, 8 authors are too much for a Review.
Anyway, the topic and the contents of the present manuscript could be interesting, but both the English language and the organization of the Review are poor and need improvement. Additionally, some minor issues need correction.
Res: Many thanks for the comments. The language of the manuscript was edited by Freescience Editorial Team. These all authors do attribute for this work, for example, literature collection, analysis, drawing, manuscript composing etc. Thus we have to list them as coauthors.
Following my opinion and suggestions.
The abstract is badly written and not clear. Please, reformulate and improve it.
Res: The abstract has been rewritten.
Introduction and related references are poor. The background provided by the authors is very limited. More information is necessary to the readers to afford the topic.
Res: "Introduction" has been polished and upgraded. The language was edited by Freescience Editorial Team. Some recent and important references have been supplemented.
As a review, the manuscript lacks completely Tables. As for the Figures, at least 5 Tables are necessary for a good work. So, authors are asked to include part of the information (the most suitable) in at least 5 Tables.
Res: Many thanks for your good advice. 2 tables have been amended in the revised manuscript. Some information displayed as the comprehensive figures is more visual and vivid than as tables. Thus, we maintain both figures and tables to present all important information for the manuscript.
A good review work should contain up-dated references. Generally, over the 50% of references should be of the last five year. No reference of 2023 is present and more that 57% of references are older than the year 2018. Please, update the references.
Res: Yes, some important and recently reported references have been supplemented.
Many Figure captions are excessively long (especially those of Figure 1, 2 and 5) and a reduction is necessary.
Res: The captions of figures have been simplified and clarified.

Reviewer 3 Report
Ni et al review the macrophages role. My concerns lie mainly in
* The reason for writing this review is not clear. Authors should write clearly the study goal.
* They should write clearly Acknowledgements, funding and conflicts of interest following the journal directions after removing them from text.
* Mpox and Covid should have separate paragraphs
* Language editing is needed.
* Language editing is needed to avoid plagiarism and errors.
Author Response
Manuscript ID: ijms-2351781
Title: Biologic mechanisms of macrophage phenotypes responding to infection and the novel therapies to moderate inflammation
Dear Dr. Kamonlak Pongpanumaporn
Assistant Editor
IJMS
Thank you very much for giving us an opportunity to revise our manuscript. We appreciate the editor and reviewers very much for their constructive comments and suggestions on our manuscript (ijms-2351781).
We have studied all comments carefully and tried to fix all the problems mentioned. According to the reviewers’ detailed suggestions, we have made a careful revision on the manuscript. All revised portions are marked in red in the revised manuscript. The language of the manuscript was edited by Freescience Editorial Team.
Reviewer 3#
Comments and Suggestions for Authors
Ni et al review the macrophages role. My concerns lie mainly in
* The reason for writing this review is not clear. Authors should write clearly the study goal.
Res: We clarified the study goal in "Abstract" and "Introduction".
* They should write clearly Acknowledgements, funding and conflicts of interest following the journal directions after removing them from text.
Res: Acknowledgements, funding and conflicts of interest were stated as the journal directions.
* Mpox and Covid should have separate paragraphs
Res: Yes, they are separately stated.
* Language editing is needed.
Res: The language of the manuscript was edited by Freescience Editorial Team.
Reviewer 4#
Comments on the Quality of English Language
* Language editing is needed to avoid plagiarism and errors.
Res: The language of the manuscript was edited by Freescience Editorial Team.
We sincerely appreciate for the work of Editors/Reviewers. If there are any other requirements or feedback, please let us know.
Thank you again
Best regards,
Yabin Zhu
On behalf of all authors,

Reviewer 4 Report
Your research was adequately performed.
Author Response
Manuscript ID: ijms-2351781
Title: Biologic mechanisms of macrophage phenotypes responding to infection and the novel therapies to moderate inflammation
Dear Dr. Kamonlak Pongpanumaporn
Assistant Editor
IJMS
Thank you very much for giving us an opportunity to revise our manuscript. We appreciate the editor and reviewers very much for their constructive comments and suggestions on our manuscript (ijms-2351781).
We have studied all comments carefully and tried to fix all the problems mentioned. According to the reviewers’ detailed suggestions, we have made a careful revision on the manuscript. All revised portions are marked in red in the revised manuscript. The language of the manuscript was edited by Freescience Editorial Team.
Reviewer 4#
Comments on the Quality of English Language
* Language editing is needed to avoid plagiarism and errors.
Res: The language of the manuscript was edited by Freescience Editorial Team.
We sincerely appreciate for the work of Editors/Reviewers. If there are any other requirements or feedback, please let us know.
Thank you again
Best regards,
Yabin Zhu
On behalf of all authors,

Round 2
Reviewer 2 Report
Dear Authors,
I appreciate your revision work. Thank you.
Author Response
Manuscript ID: ijms-2351781
Title: Biologic mechanisms of macrophage phenotypes responding to infection and the novel therapies to moderate inflammation
Point-by-point response to the comments by Editor and reviewers
Response: We appreciate reviewer 3 for the affirmation of our revised manuscript, and we would like to thank the reviewers and editor again for taking the time to review our manuscript.
We sincerely appreciate for the work of the Editors/Reviewers. If there are any other requirements or feedback, please let us know.
Thank you again
Best regards,
Yabin Zhu
On behalf of all authors,
Reviewer 3 Report
Too many sentences and paragraphs with no references to establish the information provided. It is absolutely vital.
It is really sad to find such errors after professional editing. Please rephrase ln 31-32. Additionally, according to the current CDC directions, mpox is the right term (formely monkeypox) when describing the MPX virus infection. Please keep the mpox rather the MPX virus infection.
Immunosuppression is a risk factor for every infection not only mpox. So rephrase and give more references of different infections after immunosuppression.
Correct " SARS-COVID-19"! the right term is COVID-19 epidemic rooting from SARS-CoV-2 virus.
Author Response
Manuscript ID: ijms-2351781
Title: Biologic mechanisms of macrophage phenotypes responding to infection and the novel therapies to moderate inflammation
Point-by-point response to the comments by Editor and reviewers
Reviewer 3#
Thank you very much for your professional advice. Line 31-32 was rephrased (red mark). The words, mpox & COVID-19, were corrected according to your kind reminder. Some valuable references were amended.
We know there are many reasons to cause infection, such as virus, bacteria, trauma, diabetes, and abnormal immunity etc. In this review, we focus on the biologic mechanisms of macrophages responding to infections. Virus like COVID19 was employed as an example to induce infection. Thereby, we sketch the statement of the virus and other causes.
We sincerely appreciate for the work of Editors/Reviewers. If there are any other requirements or feedback, please let us know.
Thank you again
Best regards,
Yabin Zhu
On behalf of all authors,
Round 3
Reviewer 3 Report
The study goal should be clear short if possible. Thus, authors should do their best to separate the phrase starting with "Aimimg to..." (in a separate paragraph )and rephrase it so as to be brief (even cutting it into smaller phrases).
Minor language editing is imperative.
Minor language editing is imperative.
Author Response
Manuscript ID: ijms-2351781
Title: Biologic mechanisms of macrophage phenotypes responding to infection and the novel therapies to moderate inflammation
Point-by-point response to the comments by Editor and reviewers
Reviewer 3#
Thank you very much for your professional advice. The paragraph starting with "aiming at..." was rephrased with red mark. The manuscript has been rechecked and polished with a minor revision, marked with red color.
We sincerely appreciate the work of the Editors/Reviewers. If there are any other requirements or feedback, please let us know.
Thank you again
Best regards,
Yabin Zhu
On behalf of all authors,